# Identifying Risk and Resilience Factors Impacting Mental Health among Black and Latinx Adults following Nocturnal Tornadoes in the U.S. Southeast

**DOI:** 10.3390/ijerph18168609

**Published:** 2021-08-15

**Authors:** Jennifer M. First, Kelsey Ellis, Mary Lehman Held, Florence Glass

**Affiliations:** 1College of Social Work, University of Tennessee, Knoxville, TN 37996, USA; mheld@utk.edu (M.L.H.); jglass17@vols.utk.edu (F.G.); 2Department of Geography, University of Tennessee, Knoxville, TN 37996, USA; ellis@utk.edu

**Keywords:** mental health, tornado, weather, risk communication, resilience, race

## Abstract

Prior research has found that Black and Latinx communities in the U.S. face significant disparities that impact both preparedness for severe weather events and the support received after a disaster has occurred. In the current study, we examined key risk and protective factors that impacted mental health among 221 Black and Latinx adult respondents exposed to the 2–3 March 2020 nocturnal tornado outbreak in the U.S. state of Tennessee. Key factors that adversely affected mental health among participants were encountering barriers for receiving tornado warning alerts and tornado-related exposure. Key factors that served a protective mechanism against adverse mental health included having access to physical resources, supportive relationships, and adaptive coping skills. These findings may assist National Weather Service (NWS) personnel, emergency managers, and mental health providers with the development of policies and practices to address barriers and promote protective strategies for future nocturnal tornado events.

## 1. Introduction

Extreme weather events such as severe tornadoes and storms are increasing in prevalence and intensity in the United States, particularly within the southeast areas [1]. During the overnight hours between 2–3 March 2020, 10 nocturnal tornadoes ranging from EF0–EF4 were confirmed in the U.S. state of Tennessee, killing 25 and injuring over 300 [2,3]. The physical and economic damage from the March 2020 nocturnal tornado outbreak is estimated to be between $1.5 and $2 billion and was one of the deadliest tornado outbreaks in the Middle Tennessee area [2]. Nocturnal tornadoes are 2.5 times more likely to inflict fatalities in comparison to tornadoes that occur during the daytime [4]. This is in part due to the fact that nocturnal tornadoes are more difficult to spot at night and they occur when people are sleeping and are less likely to receive warnings or other emergency information [5,6,7]. Tennessee and the broader Southeast region of the United States are at elevated risk for experiencing nocturnal tornadoes, with nearly half of Tennessee tornadoes occurring at night [5].

While direct exposure to a tornado can lead to loss of life, injury, and physical damage, exposure can also have enduring impacts on people’s mental health. Adverse mental health risks can include distress, anxiety, depression, post-traumatic stress (PTS), sleep disorders, and suicide [8,9,10]. The most prevalent mental health disorders following severe weather events are post-traumatic stress disorder (PTSD), followed by depression and anxiety [11,12]. Psychological harm can occur in the immediate aftermath of exposure and can persist over months to years [13]. For example, following a major tornado in the U.S. city of Joplin, Missouri, Houston and colleagues [14] found elevated levels of PTSD and depression for participants at both 6 months and 2.5 years following the tornado. For individuals living with adverse mental health symptoms in the aftermath of extreme weather events, a lower quality of life and functional impairments in social, occupational, and physical domains have all been reported [15].

While severe weather events may broadly affect an entire community, these challenges often disproportionately affect populations of lower economic privilege or social status [16]. Particularly in the U.S., Black and Latinx communities face significant disparities that have historically impacted both preparedness for severe weather events and the support received after a disaster has occurred. Spanish-speaking Latinxs have been shown to be less likely than English-speaking Latinx and non-Latinx whites to be prepared for potential severe weather [17], including having a plan and supplies. This disparity is not fully explained by economic disparities, and is likely exacerbated by insufficient accessibility to culturally appropriate disaster preparedness materials in languages other than English [17,18]. In addition, Black and Latinx communities face more challenges and have less access to services and resources in post-disaster settings, often due to institutional and interpersonal racism, language barriers, and distrust of governmental authorities [19,20,21].

Additional research is needed to further understand how these two groups may be impacted by nocturnal tornado events, particularly in the Southeast which has the highest tornado mortality rate in the U.S. due to many attributing factors (e.g., harder to see tornadoes due to forest cover, fewer basements and tornado shelters, high percentage of mobile homes [22]). To address this gap, we conducted a cross-sectional study among 221 Black and Latinx adults who were exposed to the 2–3 March 2020 nocturnal tornado outbreak in Middle Tennessee. Utilizing a risk and resilience framework, the current study had two aims. First, we examined (a) if participants encountered barriers to receiving tornado warning alerts during the 2–3 March 2020 nocturnal tornado outbreak, and (b) if barriers in receiving warning alerts were associated with more tornado exposure and more adverse mental health outcomes (e.g., PTS and depression). Second, we examined if resilience factors, which included physical resources, social resources, and adaptive coping skills, contributed to lower PTS and depression symptoms in participants.

### 1.1. Tornado Warnings, Exposure, and Mental Health

Various factors have been found to place individuals at increased risk for adverse outcomes following severe weather events. One such risk factor is the absence of receiving an emergency warning prior to an event. Receiving warning information, along with environmental and social cues that reinforce the presence of a hazard, is essential for encouraging the public to seek protective action during a hazardous weather event [23]. Prior research has found that issues pertaining to language, culture, and trust in public officials may inhibit the effectiveness of warning communication strategies within Black and Latinx communities [24]. For example, during Hurricane Katrina, a lack of non-English information increased disaster exposure among those with limited English proficiency and contributed to health risks after the storm and hindered recovery processes [25]. Prior research has also found that Black and Latinx residents are less likely to accept that a risk or warning alert is credible without confirmation of the alert from family or friends [26]. When tornadoes occur at night, people and their social networks are more likely to be sleeping and depending on their physical resources (e.g., no access to a smart phone to receive warnings, no access to NOAA radio), they are less likely to receive warnings [6]. An absence of receiving an effective emergency warning not only places an individual at elevated risk for physical injury, but may also contribute to stress-induced trauma of the storm hitting while being unprepared [27,28,29].

In addition, prior studies have found that being exposed to severe weather-related experiences (e.g., having property damage, losing a loved one, being injured, fearing for one’s life) increases one’s risk for adverse mental health outcomes (for a review see Neria et al. [30]). Empirical studies investigating the relationship between disaster exposure and mental health have found a dose-response effect, in which PTS and depression symptoms are found to increase with greater levels of exposure or stressful experiences related to the disaster event [30,31,32,33,34]. For instance, following a major tornado in Joplin, Missouri, Houston et al. [14] found having more tornado-related exposure (e.g., hearing and seeing the tornado, property damages, injuries) was related to a greater likelihood of PTSD and depression for participants. In terms of racial and ethnic disparities, previous research found Black and Latinx participants encountered high levels of disaster exposure following Hurricane Ike, which contributed to PTSD and depression [35]. In the current study, we predict that encountering barriers to receiving tornado warning alerts will be associated with more tornado exposure in participants. We also predict that barriers to receiving tornado warning alerts and tornado exposure will be associated with more adverse mental health.

**Hypothesis** **1** **(H1).**
*More barriers to receiving tornado warning alerts will be associated with more tornado exposure.*


**Hypothesis** **2** **(H2).**
*(a) More barriers to receiving tornado warning alerts, and (b) more tornado exposure, will be associated with more PTS and depression symptoms.*


### 1.2. Resilience Factors and Mental Health

Finally, from a protective perspective, prior research has suggested resilience is a process of harnessing resources to sustain well-being in the face of difficulties [36,37]. Within Black and Latinx communities, prior research found resilience may be promoted through physical resources, social support, optimism, and cultural pride as a means of coping with stressors and promoting psychological well-being [38,39,40,41,42,43]. Disaster mental health scholars have been successful in identifying internal personality traits that support psychological resilience. For instance, Osofsky and colleagues [44] found that more self-efficacy was associated with lower psychopathologies (e.g., depression, PTSD) for individuals exposed to both Hurricane Katrina and the Deepwater Horizon oil spill. However, additional studies are needed that take into account the socio-ecological factors (e.g., housing and economic stability, insurance, community assistance, cultural and social networks) that may contribute to post-disaster resilience in diverse populations [40,45,46,47,48,49]. Therefore, we predict in the current study, resilience factors that include socio-ecological and psychological resources [46] will have a significant and inverse association with PTS and depression symptoms among participants.

**Hypothesis** **3** **(H3).**
*Higher level of resilience (e.g., access to physical resources, social support, and adaptive coping skills) will contribute to lower PTS and depression symptoms.*


## 2. Materials and Methods

This study uses a sample of Black and Latinx adults (*N* = 221) from Middle Tennessee to examine if weather alert barriers encountered during nocturnal tornadoes are related to more tornado exposure and adverse mental health, and if individual-level resilience helps protect against posttraumatic stress (PTS) and depression symptoms. We utilized Structural Equation Modeling (SEM) to test hypothesized associations between barriers to tornado warning alerts, tornado exposure, resilience, PTS, and depression.

### 2.1. Participants and Procedures

Data collection procedures were approved by the University of Tennessee Institutional Review Board (IRB). Data were collected via an online survey conducted in April 2021, approximately one year following the March 2020 tornadoes. The sample included 221 adults (18 or older) residing in a Middle Tennessee county (Benton, Humphreys, Davidson, Wilson, Smith, Putnum, and Cumberland) impacted by the 2–3 March nocturnal tornadoes. The survey was conducted via Qualtrics Panels which recruits from a pool of U.S. adults to participate in an online research panel via the company. The company is able to recruit participants in targeted areas and provides participants with compensation through Qualtrics incentive program, which includes prize drawings and accumulated rewards for participants. To participate in the study, individuals were required to be 18 years or older, identify as Black or Latinx, speak English or Spanish, and have access to the internet. Potential respondents were sent an email invitation with a secure URL from Qualtrics to access the survey and review the study’s purpose. Participants first read a consent form and were required to provide their consent to participate in the study by selecting an “I agree to participate” button. After consenting to the study, participants were directed to the online survey.

### 2.2. Measures

Barriers to Tornado Warning Alerts: The survey included questions that were developed to understand potential barriers to receiving tornado warnings. Participants answered no (0) or yes (1) related to six barriers to warnings (M = 1.45, SD = 1.05). Barriers to warnings questions asked participants if they encountered any of the following barriers: encountering language barriers, being asleep, no access to a smart phone to receive alerts, no access to a NOAA weather radio to receive alerts, family, friends and neighbors were asleep, not hearing sirens. The scores of all items were summed to create an observed variable. The scores ranged from 0 to 6, with a higher score indicating more barriers to receiving warning information.

Tornado Exposure: The survey included questions that were developed to understand participants’ exposure to the 2–3 March 2020 tornadoes. Participants answered no (0) or yes (1) related to six exposure items (M = 3.89, SD = 1.54) adapted from prior studies to assess tornado-related stressful experiences [15]. Tornado exposure questions asked participants if they had their property damaged, experienced injury, had family or friends with property damage, experienced feelings of helplessness or fear, believed they or someone they knew would be killed or harmed by the tornado, and saw scenes of aftermath and damaged areas. The scores of all items were summed to create an observed variable. The scores ranged from 0 to 6, with a higher score indicating more tornado exposure.

Resilience: Resilience (M = 110.55, SD = 36.45) was measured via the Disaster Adaptation and Resilience Scale [46], a 43-item multidimensional scale designed to measure socio-ecological and psychological protective factors supporting adult resilience in disaster contexts. The scale consists of five domains found to support individual resilience, including: physical resources, social resources, distress regulation, problem-solving, and optimism. Sample items include “I have stable or permanent housing,” “I have access to reliable transportation,” “I have a safe place to go in the event of a disaster” (physical resources); “I have people I can turn to and ask for help” (social resources); “I give myself time to recover from upsetting situations” (distress regulation); “I look for information or resources to help deal with challenges” (problem-solving); and “I believe I will make it through difficult times” (optimism). Each item is rated on a 5-point Likert scale ranging from 0 (not at all true) to 4 (true nearly all of the time), with higher scores reflecting higher levels of resilience. In the present sample, Cronbach’s alpha value was 0.97.

Post-traumatic stress: PTS symptoms (M = 36.32, SD = 18.26) were measured with the Post-traumatic Stress Disorder Checklist for Civilians (PCL-C) [50], a 17-item self-report questionnaire that assesses for probable PTSD diagnosis in individuals exposed to a traumatic event. The PCL-C has four subscales, including re-experiencing symptoms, avoidance symptoms, negative alterations in cognition and mood and arousal symptoms. Each item is scored on a five-point Likert scale ranging from not at all (1) to extremely (5). Respondents were asked to indicate how often they were bothered by each of the symptoms during the past month related to the March 2020 tornadoes. In the present sample, Cronbach’s alpha value was 0.96.

Depression: Symptoms of depression (M = 7.20, SD = 7.44) were assessed with the Patient Health Questionnaire-9 (PHQ-9) [51]. The PHQ measures the degree to which an individual has experienced depressed mood and anhedonia over the past two weeks in order to screen participants for depression. Respondents were asked to indicate how often they were bothered by each symptom in the past two weeks using four response options ranging from not at all (0) to nearly every day (3), and whether the symptoms endorsed occurred within the same two-week period. In the present sample, Cronbach’s alpha value was 0.93.

### 2.3. Analyses

Data analysis was conducted using R statistical software and packages [52]. Descriptive statistics of respondents’ demographics and variables of interest were analyzed using univariate methods including means, standard deviations, frequencies, and percentages as appropriate. To examine the relationships between risk and protective factors, we used structural equation modeling (SEM) to test our hypothesized relationships. SEM has two important advantages for this study’s analysis. First, SEM is able to estimate latent variables from their indicators, rather than summed variables from the average of scale items [53]. Thus, measurement error is essentially eliminated and analysis estimates represent the true scores of latent relationships. Second, SEM allows for the testing of complex relationships between observed and latent variables simultaneously to test theories of causal relationships [53].

Using a two-step procedure recommended by Kline [53], we first conducted a confirmatory factor analysis (CFA) to establish that the latent variables (e.g., resilience, PTS, depression) were well explained by the indicators using confirmatory factor analysis (e.g., λ > 0.50). To obtain standardized, unit-free estimates that reflect the indicator reliabilities, the scale was set by the fixed factor method, which fixes the latent variance to one (e.g., ψ = 1.0) and we used a robust maximum likelihood estimation to ensure multivariate normality. In cases of missing data, a full information maximum likelihood (FIML) estimation was implemented, which assumes missing data points have an expectation equal to a model-derived value estimated from the remaining data points. To evaluate construct validity of measures, the average variance extracted (AVE) was calculated based on the CFA model. AVE values greater than 0.50 were deemed acceptable. After establishing the measurement model, we estimated a structural model to conduct a path analysis of risk and protective factors on mental health outcomes. To examine model fit, we used Little’s [54] guidelines for goodness of fit indices, including root mean square error of approximation (RMSEA; values of 0.08 or less indicate adequate fit), standardized root mean square residual (SRMR; values of 0.08 or less indicate adequate fit), Tucker–Lewis index (TLI; which should be equal to, or greater than, 0.95), and comparative fit index (CFI; which should be equal to, or greater than, 0.95).

## 3. Results

Missing data in the current study did not exceed 5% for any variable. Of the 221 participants, 150 were female (67.9%) 68 were male (30.8%), 2 individuals identified as transgender (0.90%), and 1 individual as non-binary (0.50%). Participants identified as Black (n = 150, 67.9%) or Hispanic/Latino (n = 71, 32.2%). The age of participants ranged with 18–29 years old at 56.6% (n = 125), 30–49 years old at 30.8% (n = 68), 50–69 years old at 10.9% (n = 24), and 70 years or older at 1.8% (n = 4). The majority of participants at the time of the study lived in a housing structure that was a house detached from other buildings (45.9%), followed by an apartment building (21.7%), a house attached to other buildings (17.3%), or a mobile home (14.1%). Descriptive statistics found participants encountered a variety of barriers to receiving tornado warning alerts. These included being asleep (47.1%), followed by not hearing tornado sirens (27.6%), family, friends, neighbors were asleep (26.2%), no access to a smart phone to receive weather emergency alerts (18.6%), no access to a NOAA weather radio to receive alerts (14.9%), and encountering language barriers (11.3%). See Table 1 for descriptive statistics for demographics and variables of interest.

Next, we used structural equation modeling (SEM) to identify the relationships between risk (i.e., barriers to warnings, tornado exposure) and resilience factors on mental health outcomes. Our initial SEM measurement model converged and all factor loadings for each latent variable showed acceptable level with the λ values above 0.50. However, the CFA model revealed unacceptable levels of goodness of fit (i.e., both CFI and TLI were less than 0.95) due to a high number of indicators (43 items) for the resilience latent variable. To remedy this problem, the 43 resilience items were fit into five parcels so that each parcel formed a theoretically meaningful cluster related to the five factors of the scale [54]. The resilience factor parcels showed acceptable to high factor loadings (0.515–0.925) on the resilience latent variable, indicating they represented the latent variable well. After parceling the resilience variable, the measurement model exhibited acceptable fit with the data and the model fit statistics were: χ2(116) = 147.031, *p* < 0.01, CFI = 0.986, TLI = 0.984, RMSEA = 0.039, SRMR = 0.039. Based on the CFA model, the AVE values were calculated and were all found to be above 0.50 indicating construct validity for each of the measures. After establishing the measurement model, we estimated the structural relationships between the observed and latent variables. The structural model achieved acceptable fit, model fit statistics were: Model Fit: χ2(146) = 206.495, *p* < 0.01, CFI = 0.975, TLI = 0.970, RMSEA = 0.048, SRMR = 0.051 and allowed for the testing of our hypotheses.

Our first hypothesis (H1) predicted that encountering more barriers to receiving tornado warning alerts would be associated with more tornado exposure. H1 was supported, as results found barriers to tornado warning alerts had a significant and positive relationship with more tornado exposure (β = 0.196, *p* < 0.05). Next, our second hypothesis (H2a) predicted more barriers to receiving tornado warning alerts would be associated with more PTS and depression symptoms. H2a was confirmed as results found more barriers to warning alerts was related to PTS (β = 0.200, *p* < 0.01) and depression symptoms (β = 0.268, *p* < 0.001). Our second hypothesis (H2b) predicted more tornado exposure would have a significant and positive relationship with PTS and depression symptoms. H2b was also confirmed as results found more tornado exposure had a significant and positive relationship with PTS (β = 0.443, *p* < 0.001) and depression symptoms (β = 0.427, *p* < 0.001). Finally, our third hypothesis (H3) predicted that higher levels of resilience would have a significant and negative relationship with PTS and depression symptoms. H3 was confirmed as resilience had a significant and inverse relationship with PTS (β = −0.149, *p* < 0.01) and depression symptoms (β = −0.229, *p* < 0.001). See Table 2 and Figure 1 for the structural results.

## 4. Discussion

In the current study we conducted a survey with Black and Latinx adult participants in Middle Tennessee who were exposed to a nocturnal tornado outbreak to examine risk and resilience factors that impacted mental health outcomes. Our results point to several main findings. First, we identified various barriers encountered by Black and Latinx adults for receiving emergency alert information during the 2–3 March 2020 nocturnal tornado outbreak. Respondents reported barriers that included being asleep, having family, friends, and neighbors asleep, encountering language barriers, no access to a smart phone to receive alerts, no access to a NOAA weather radio to receive alerts, and not hearing sirens. These findings are consistent with prior research using hypothetical scenarios that showed many people do not believe they would get a tornado warning should one be issued at night [7] due to factors like deep sleep or unreliable Wireless Emergency Alerts [55]. At night, individuals sleeping in mobile home structures are particularly in danger should they not be awoken, as tornado fatalities in manufactured homes in Southeast occur disproportionately at night [56].

Second, Black and Latinx respondents who reported barriers to receiving nocturnal tornado warning alerts were more likely to report tornado exposure (i.e., injury, fearing for one’s life) and adverse mental health outcomes (e.g., PTS and depression). Previous severe weather studies have shown depression and PTS symptoms are largely predicted by the amount of exposure or experiences related to severe weather events [30,32]. The current study illustrates that encountering barriers to receiving nocturnal tornado warning alerts predicted more tornado exposure, which in turn was associated with more PTS and depression symptoms. This finding is consistent with research following hurricane Katrina which found Black and Latinx individuals encountered more barriers for receiving emergency information which contributed to more hurricane exposure and adverse mental health [18,25].

Third, consistent with other research [44,57] we found an inverse relationship between resilience and adverse mental health among participants. Specifically, we found that resilience factors consisting of external physical resources, social resources, and adaptive coping strategies, served a protective function for respondents and were associated with lower levels of PTS and depression symptoms. The measure we used in the present study consisted of five dimensions (physical resources, social resources, problem-solving, distress regulation, and optimism) and was adapted specifically for use related to individual resilience following disasters [46]. This approach is distinct from previous disaster resilience studies which have primarily focused on measuring psychological facets or individual personality traits [58]. While general measures of resilience are useful in measuring psychological traits associated with resilience, these traits represent only the internal adaptive capacities and neglect broader socio-ecological resources that are needed for human adaptation [36,49]. The current study builds upon prior studies [40,45,59,60] that have highlighted the importance of socio-ecological resources being available following extreme weather events and the need for examining pathways of resilience in the face of disaster, particularly among African American and Latinx survivors [35]. However, structural inequalities and injustices limit access to key resources necessary for resilience, further compounding trauma and increasing risk for adverse mental health [24]. While our measure of individual-level resilience included assessment of socio-ecological and psychological resources, (i.e., “I have enough food to eat, I have stable housing, I have access to reliable transportation, I have a safe place to go in the event of a disaster, I am treated fairly by people in my community, I believe I will make it through difficult times”), future longitudinal studies are needed to examine how macro-level forces shape the distribution of these resources across racial and ethnic groups.

Finally, it is important to note that just eight days following the March 2020 tornado outbreak, the coronavirus infectious disease outbreak was declared a global pandemic by the World Health Organization. Given the disproportionate burden of the COVID-19 pandemic on Black and Latinx communities [61], pandemic-related stressors may have further compounded PTS and depression symptoms among respondents. Future research should examine the potential cumulative mental health impacts of exposure to severe weather events during the COVID-19 pandemic. While we did not focus on pandemic-related stressors, given the timing of the tornado outbreak and the pandemic, our resilience findings provide important insights into protective factors that were associated with lower levels of PTS and depression symptoms for Black and Latinx respondents.

### 4.1. Implications

In terms of implications, the current study’s findings can assist National Weather Service (NWS) personnel, emergency managers, and mental health providers with the development of policies and practices to strengthen preparedness and warning systems to aid in overall safety and well-being of socially vulnerable communities. National Weather Service offices and emergency management professionals should build partnerships with Black and Latinx leaders, organizations, and residents to develop equitable policies and practices to address barriers to tornado warnings, prior to and during an event [62]. Use of social media and flyers to educate vulnerable communities about nocturnal tornado risks and preparedness strategies could strengthen knowledge on the front end that promotes safety and well-being [63]. Culturally and linguistically competent mental health and social service provider organizations can also serve a vital role in preparedness and recovery [64]. At a direct practice level, providers situated in high-risk communities can include initial assessment questions inquiring about safe sheltering options and warning alert access as significant factors contributing to health and well-being outcomes.

In addition, a central piece for promoting resilience before and after disaster events is identifying the factors and resources that have been found to protect individuals from negative outcomes following disaster-related adversity. This study found that Black and Latinx participants with more physical, social, and psychological resources were better equipped to be protected from adverse mental health outcomes following disaster events. Mental health intervention efforts should focus on mitigating disaster distress by advocating for policies and practices that eliminate inequalities and mobilize physical, social, and psychological resources.

### 4.2. Limitations

As with all research studies, this study had several limitations. First, this study utilized an online questionnaire which required internet access and may have limited participation among those with lower incomes, less education, and of older age. Second, this study utilized self-report measures which may not be as accurate as a full clinical evaluation of PTS and depression symptomatology. Third, participants were exposed to the COVID-19 pandemic which may have further compounded PTS and depression symptoms. Fourth, this study was cross-sectional in design and therefore the collected data precludes causal claims of temporal order [65]. However, the present study presents a model that is grounded in the theoretical literature and was supported by previous research investigations [6,7,20], all of which provide a compelling case for investigating the relationships we conducted in the current study. Despite these limitations, this study takes an important step towards identifying and testing risk and protective factors to identify how they contribute to mental health outcomes in Black and Latinx populations exposed to a tornado outbreak.

## 5. Conclusions

In the current study, we examined key risk and protective factors that impacted mental health among Black and Latinx adult respondents exposed to a nocturnal tornado outbreak in the U.S. state of Tennessee. We found participants identified a variety of barriers in accessing emergency warning alerts during nocturnal tornadoes and that encountering barriers to tornado warnings were associated with more tornado exposure and more PTS and depression. We also found that higher levels of resilience (physical, social, and psychological resources) had an inverse relationship with adverse mental health outcomes among respondents. These findings are relevant to weather personnel, emergency managers, and mental health providers to aid in overall safety and well-being of socially vulnerable communities to future nocturnal tornado threats.

## Figures and Tables

**Figure 1 ijerph-18-08609-f001:**
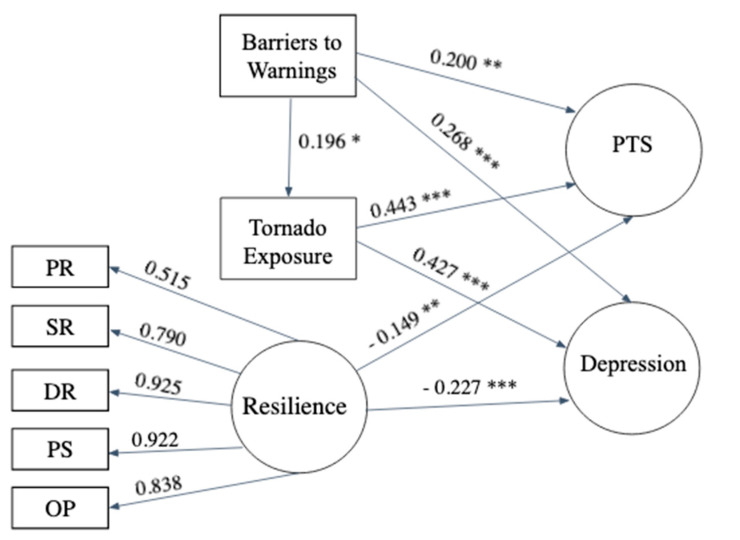
Diagram of Structural Model. *Note:* Model Fit statistics: Model Fit: χ2(146) = 206.495, *p* < 0.01, CFI = 0.975, TLI = 0.970, RMSEA = 0.048, SRMR = 0.051. PR = Physical Resources, SR = Social Resources, DR = Distress Regulation, PS = Problem Solving, OP = Optimism, PTS = Posttraumatic stress symptoms. ** p* < 0.05, ** *p* < 0.01, *** *p* < 0.001.

**Table 1 ijerph-18-08609-t001:** Descriptive Statistics.

Variables	*N*	%
Gender		
Female	150	67.9
Male	68	30.8
Transgender	2	0.90
Non-binary	1	0.50
Race/Ethnicity		
Black/African American/Afro-Caribbean	150	67.9
Hispanic/Latino	71	32.2
Age		
18–29	125	56.6
30–49	68	30.8
50–69	24	10.9
Over 70	4	1.8
Income		
Less than $15,000	47	21.5
$15,000 to $29,999	34	15.5
$30,000 to $44,999	40	18.3
$45,000 to $59,999	32	14.6
$60,000 to $74,999	20	9.1
$75,000 to $104,999	24	11.0
$105,000 to $119,000	22	10.0
Education		
Grade School	4	1.8
Some High School	16	7.2
High School Graduate	63	28.5
Some College	59	26.7
College Graduate	52	23.5
Advanced Degree	27	12.2
Housing Structure		
Mobile Home	31	14.1
House detached from other buildings	101	45.9
House attached to other buildings	38	17.3
Apartment building	48	21.7
Boat, RV, Van, etc.	2	0.90
Barriers to Tornado Warning Alerts		
Language barriers	25	11.3
Being asleep	104	47.1
No access to smart phone for alerts	41	18.6
No access to NOAA weather radio	33	14.9
Family, friends, neighbors were asleep	58	26.2
Did not hear sirens	61	27.6
Tornado Exposure		
Property damage in tornado	87	39.4
Injured from tornado	35	6.31
Knew people with property damage from tornado	146	66.0
Believed self or loved one would be killed or harmed by tornado	149	67.4
Felt helplessness, fear, or horror during tornado	153	69.2
Viewed damaged areas, debris, people injured after tornado	175	79.2

**Table 2 ijerph-18-08609-t002:** Structural model: regression paths.

Regression Paths	Unstandardized Estimate	Standard Error	Standard Estimate
Tornado Exposure (R^2^ = 0.038)			
Barriers to Warnings	0.238	0.032	0.196 *
Post-traumatic Stress (R^2^ = 0.293)			
Barriers to Warnings	0.224	0.065	0.199 **
Tornado Exposure	0.117	0.022	0.443 ***
Resilience	−0.177	0.081	−0.149 **
Depression (R^2^ = 0.350)			
Barriers to Warnings	0.315	0.074	0.268 ***
Tornado Exposure	0.118	0.019	0.427 ***
Resilience	−0.285	0.068	−0.229 ***

*Note:* Model Fit statistics: Model Fit: χ2(146) = 206.495, *p* < 0.01, CFI = 0.975, TLI = 0.970, RMSEA = 0.048, SRMR = 0.051. ** p* < 0.05, ** *p* < 0.01, *** *p* < 0.001.

## Data Availability

Data is available by requesting access from the corresponding author J.M.F.

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
