# Peer review of "Identifying Risk and Resilience Factors Impacting Mental Health among Black and Latinx Adults following Nocturnal Tornadoes in the U.S. Southeast"

_ijerph, 2021, doi:10.3390/ijerph18168609_

Round 1
Reviewer 1 Report
This study examined risk and resilience/protective factors in Black and Latinx adults who were exposed to a nocturnal tornado outbreak in March 2020. The manuscript was clear, well-written, and organized. The study has many strengths, including providing much-needed clarity on specific factors that contribute to adverse outcomes (and resilience) in communities who have been historically disinvested and are more at-risk to the negative consequences of extreme weather events. Slight alterations and some specific requests for clarification are provided to improve the manuscript. As a whole, this was an important study with results that can provide useful recommendations for disaster preparedness efforts.
Introduction
- p. 2, lines 46-48: It would be helpful if the authors could specify whether the "adverse mental health symptoms" are the same as those previously mentioned (i.e., those that arise in the aftermath of EWEs) or if the symptoms are referring to pre-existing mental health conditions/symptoms. I believe the authors are referring to the same symptoms they had been discussing, in which case it could be helpful to say something like, "For individuals living with adverse mental health symptoms in the aftermath of extreme weather events, ...."
- p. 2, line 57: The authors may wish to use the word "exacerbated" instead of "bolstered" to describe the disparity.
- p. 2, line 64: It would be helpful to define physical factors (or give an example) for readers who may not be familiar. The authors provide helpful examples of physical factors later in the manuscript. I recommend providing those examples in line 64 rather than later on.
- p. 2, lines 84-86: The sentence is somewhat confusing for the reader. The authors may wish to replace the word "is" with "as" or to insert the word "that" after the word "accept."
Materials and Methods
5. p. 3, 2.1. Participants and Procedures: I interpreted from the study description and funding acknowledgement that participants completed the survey on a completely volunteer basis and were not paid for their time by either the study's authors or by Qualtrics Panels. This detail would be helpful to clarify in the "Participants and Procedures" section. The authors may also consider briefly adding this information to the first limitation in the Limitations section, as a group of individuals who volunteer their time may have different characteristics than those who are paid for their time.
6. p. 4, 2.2. Measures, lines 199-201: The description of depression symptom measurement is somewhat confusing from an assessment perspective and needs clarification. There are two particularly confusing aspects: 1) Please expand on what is meant by "bothered by each symptom related to the March 2020 tornadoes." For example, are the authors saying that symptoms were only endorsed if the respondent considered that symptom to be related to the tornadoes (and not anything else)? This wording makes good sense for assessment of PTS symptoms because the tornado outbreak would be the index trauma; however, this wording makes less sense when assessing depression symptoms. 2) Please clarify the time period in which participants were reporting on their depression symptoms. For example, could they have experienced the depressive symptom at any time since March 2020? In general, more detail regarding how depression symptoms were measured is needed, as depressive symptoms are a latent variable and clarification on this matter is central to understanding the results. Further, depending on how depressive symptoms were measured, this will likely need to be considered in the limitations section (particularly in light of the timing of the pandemic and the fact that we have seen increased anxiety/depressive symptoms in general during the time between the tornado outbreak and when the Qualtrics survey was completed).
Results
Overall, results were presented clearly and were well-written.
7. p. 5, lines 235-238: Please clarify if participants were living in these housing arrangements at the time of the tornado outbreak or at the time they completed the survey.
8. Table 2: Small visual edit -- the word "resilience" is not aligned with other factors associated with depressive symptoms
Discussion:
The discussion was well-written and clear. The implications section was particularly strong and provided useful, actionable suggestions.
9. In the discussion, the authors should explore the possible impacts of the pandemic on participants' responses, as participants were asked to describe risk and resilience factors that may have been greatly affected by the pandemic (e.g., social resources, optimism). The authors may also wish to include a few ways in which the pandemic may have influenced variables of interest in the aftermath of the tornado, which temporally aligned with the beginning of the global pandemic (for example, that both the pandemic and extreme weather events affect(ed) Black and Latinx communities disproportionally and may have had exacerbating reciprocal influences on one another). The findings on resiliency and protective factors seem especially impactful in light of the timing of the tornado outbreak and pandemic.
Conclusions:
The conclusions section was well-written and clear.
Reviewer 2 Report
The work is original, interesting, and surely has different merits. However, the validation of the models should be enlarged, clarified, and improved, adding some explanation on why the authors to conclude on validating their findings. The authors can also improve the explanations about the data quality of their study (presence of missing data, for instance). Of course, "data quality" does not mean a work loses interest if exists some problems. Still, it is relevant to how these eventual problems are handled (for instance, many different approaches to handling missing data). Overall, Nevertheless, the work is auspicious.
Author Response
Response: Thank you for this important question. We have expanded our description of the procedures on validation of our models (see pages 5-7). We have also included an explanation of how we dealt with missing data (p. 5) - a key step we omitted from the original manuscript.